# Spectral estimation for detecting low-dimensional structure in networks using arbitrary null models

**Mark D. Humphries**[1,2]*, **Javier A. Caballero**[2,3], **Mat Evans**[1,2], **Silvia Maggi**[1,2], **Abhinav Singh**[1]

**1** School of Psychology, University of Nottingham, Nottingham, United Kingdom, **2** Faculty of Biology, Medicine, and Health, University of Manchester, Manchester, United Kingdom, **3** Department of Psychology, University of Sheffield, Sheffield, United Kingdom

☯ These authors contributed equally to this work.
¤ Current address: Department of Automatic Control and Systems Engineering, University of Sheffield, Sheffield, United Kingdom
* mark.humphries@nottingham.ac.uk

**Data Availability Statement:** A full list of URLs to all code and data are provided in the dedicated "Data and code availability" section of the Methods. All code and data necessary to reproduce the

## Abstract

Discovering low-dimensional structure in real-world networks requires a suitable null model that defines the absence of meaningful structure. Here we introduce a spectral approach for detecting a network's low-dimensional structure, and the nodes that participate in it, using any null model. We use generative models to estimate the expected eigenvalue distribution under a specified null model, and then detect where the data network's eigenspectra exceed the estimated bounds. On synthetic networks, this spectral estimation approach cleanly detects transitions between random and community structure, recovers the number and membership of communities, and removes noise nodes. On real networks spectral estimation finds either a significant fraction of noise nodes or no departure from a null model, in stark contrast to traditional community detection methods. Across all analyses, we find the choice of null model can strongly alter conclusions about the presence of network structure. Our spectral estimation approach is therefore a promising basis for detecting low-dimensional structure in real-world networks, or lack thereof.

## Introduction

Network science has given us a powerful toolbox with which to describe real-world systems, encapsulating a system's entities as $n$ nodes and the interactions between them as links. But the number of nodes is typically between $10^2$ and $10^9$ [1, 2], so networks are inherently high-dimensional objects. For a deeper understanding of a network constructed from data, we'd like to know if that network has some simpler underlying principles, some kind of low-dimensional structure. Two questions arise: what is that low-dimensional structure, and which nodes are participating in it? We propose here a spectral approach to answer both questions.

One way of detecting low-dimensional structure is to specify a null model for the absence of that structure, then detect the extent to which the data network departs from that null

figures in the paper are hosted at: https://github.com/mdhumphries/NetworkNoiseRejection.

**Funding:** This work was supported by grants to M. H. from the Medical Research Council [grant numbers MR/J008648/1, MR/P005659/1, and MR/S025944/1] and a National Council of Science and Technology (CONACyT) Fellowship to J.C. The funders had no role in study design, data collection and analysis, decision to publish, or preparation of the manuscript.

**Competing interests:** The authors have declared that no competing interests exist.

model. The problem of community detection is a prime example of this approach, where we seek to determine whether a network contains groups of nodes that are densely interconnected [3–7]. To give some examples [6, 8, 9]: in social networks, these communities may be groups of friends, of collaborating scientists, or of people with similar interests; in biological networks these groups may be interacting brain regions, interacting proteins, or interacting species in a food web; more abstractly, these groups may be webpages on the same topic, or words commonly found together in English. Finding such a community structure in a data network requires a null model against which to assess the relative density of connections in the data. We use community detection as our prime application here as it illustrates both the key challenges we want to address.

First, community detection algorithms overwhelmingly use the same null model, the expectation of the configuration model [4, 6], which then often determines the form of the algorithm itself. Moreover, typically community detection algorithms are unable to detect the absence of community structure. We'd like the freedom to choose a null model for how we want to define a community structure, or its absence, not least different variants of the configuration model itself [10]. This is an example of the more general problem of detecting low-dimensional structure in a network, for which we would like to able to use any suitable null model for that structure.

Second, community detection algorithms rarely consider the problem of nodes that do not belong to any community [11, 12]. In any network constructed from data, such "noise" nodes may be present due to sampling error [11], or to sparse sampling of the true network (as in networks of connections between neurons). Or they may be generated by some random process marginally related to the construction of the main network, such as minor characters in narrative texts. However they arise, ideally we would be able to detect such noise nodes by defining them against the same null model for the absence of communities. This is a special case of the more general problem of detecting nodes that are not participating in the low-dimensional structure of a network.

Our proposed solution to these challenges is a simple sampling approach. We frame the departure between data and model as a comparison between the eigenspectrum of a real-world network and that predicted by the specified null model, an approach motivated by current spectral approaches to community detection [4, 13–17]. We give an algorithm that samples networks from a generative null model to determine the expected bounds on the data network's eigenspectra if it was a sample from that generative model. The upper bound is used to construct a low-dimensional projection of the data network from its excess eigenvectors; we then use this projection to reject nodes that do not contribute to these dimensions, extracting a "signal" network. All code for this framework is provided at https://github.com/mdhumphries/NetworkNoiseRejection.

Applying our spectral estimation approach to synthetic networks with planted communities, we show that the low-dimensional projection recovers the correct number of planted communities, and successfully rejects noise nodes around the planted communities. On real-world networks, we show significant advantages over community detection alone, which finds community structure in every tested network. For example, our spectral approach reports no community structure in the large co-author network of the Computational and Systems Neuroscience (COSYNE) conference, pointing to a lack of disciplinary boundaries in this research field. We also demonstrate that spectral estimation can recover $k$-partite structure in a sub-set of real networks. Finally, we show that the choice of null model can strongly alter conclusions about the low-dimensional structure in both synthetic and data networks. Our spectral estimation approach is a starting point for developing richer comparisons of real-world networks with suitable generative models.

## Results

Our goal is to compare a weighted, undirected data network $\mathbf{W}$ to some chosen null model that specifies the absence of a particular low-dimensional structure. A simple way to compare data and null models is $\mathbf{C} = \mathbf{W} - \langle\mathbf{P}\rangle$, where the matrix $\langle\mathbf{P}\rangle$ is the expected weights under some null model. For example, if we choose $\mathbf{P}$ to be the classic configuration model, then $\mathbf{C}$ is the modularity matrix [4]; as we are framing this as a general problem, we thus here term $\mathbf{C}$ the comparison matrix. Our idea is to test whether the data network is consistent with being a realisation of the generative process whose expectation is $\langle\mathbf{P}\rangle$, namely that $\mathbf{W} \approx \langle\mathbf{P}\rangle$.

To do so, we generate a set of sample null model networks $\mathbf{P}_1^*, \ldots, \mathbf{P}_N^*$ from the generative model whose expectation is $\langle\mathbf{P}\rangle$. We then compute each sample's comparison matrix $\mathbf{C}_i^* = \mathbf{P}_i^* - \langle\mathbf{P}\rangle$, and the eigenspectra of $\mathbf{C}_i^*$. Combining the sampled eigenspectra thus estimates the expected eigenspectrum of $\mathbf{C}$ due solely to variations in the null model, illustrated schematically in Fig 1a. Here we focus on estimating its upper bound: finding eigenvalues of the data network's comparison matrix exceeding that bound is then evidence of some low-dimensional structure in the data that departs from the null model. Moreover, the number of eigenvalues exceeding the upper bound gives us an estimate of the number of dimensions of that low-dimensional structure.

We can then obtain a low-dimensional projection of the data network (Fig 1b), by using the eigenvectors of $\mathbf{C}$ corresponding to the eigenvalues that exceed the limits predicted by the model. In that low-dimensional projection, we can do two things: first, test if individual nodes exceed the predictions of the null model, and reject them if not (Fig 1c, grey circles); second, cluster the remaining nodes (Fig 1c, coloured circles). Full details of this spectral estimation process are given in the Methods.

The choice of null model network is limited only to those which can be captured in a generative process, for we need to sample networks. For some null models, like the classic configuration model, we know $\langle\mathbf{P}\rangle$ explicitly; if not, we can simply estimate $\langle\mathbf{P}\rangle$ as the expectation over the sampled networks. We use two null models here. One is the weighted version of the classic configuration model (WCM) [18]. For the second we introduce a sparse variant (sparse WCM) that more accurately captures the distribution of weights in the data network, as we illustrate for a real network in the Fig 1 in S1 Appendix. As we show below, the choice of null model is crucial.

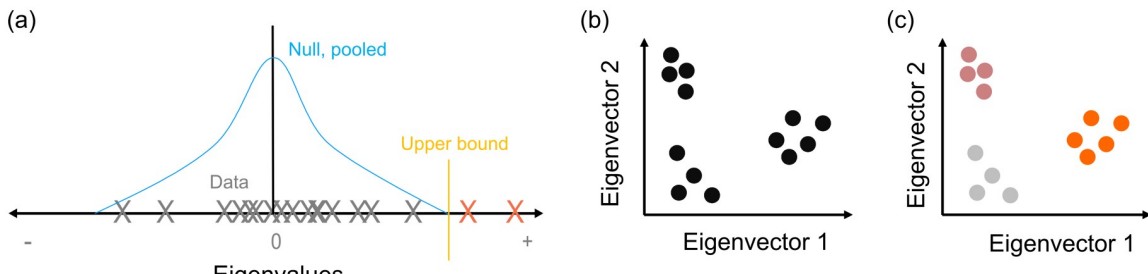

**Fig 1. Elements of the spectral estimation algorithm.** (a) Schematic of eigenvalue spectra. We estimate the null model's distribution of eigenvalues (blue) for $\mathbf{C}$, by generating a sample of null model networks. The vertical orange line is the estimate of the distribution's upper bound. The eigenvalues of the data network (crosses) are compared to the upper bound; those above the upper bound (red) indicate low-dimensional structure departing from the null model. (b) Schematic of low-dimensional projections of the network after spectral estimation. Retained eigenvectors, corresponding to eigenvalues above the null model's upper bound, define a projection of the network's nodes (circles). (c) Schematic of node rejection. Nodes close to the origin (grey) do not contribute to the low-dimensional structure of the network, so are candidates for rejection. Nodes far from the origin are contributing, potentially as clusters (colours).

## Spectral estimation detects transitions to community structure

We first ask if our spectral estimation approach is able to correctly detect networks with no low-dimensional structure. As our application here is community detection, we construct synthetic weighted networks with planted communities. Each synthetic network has $n = 400$ nodes divided into $q = 4$ equal-sized groups, and its adjacency matrix $\mathbf{A}$ is constructed by creating links between groups with probability $P$(between) and within groups with probability $P$ (within). The corresponding weight matrix $\mathbf{W}$ is then constructed by assigning sampled weights to those links (see Methods). By increasing the difference $P$(within) − $P$(between) from zero, we move from a random weighted network to a strongly modular weighted network.

Fig 2a shows that spectral estimation can consistently identify the absence of modular structure in synthetic networks when none is present, and transitions sharply to consistently detecting modular structure as the synthetic networks depart from random. Crucially, correct performance depends on the choice of null model: using the sparse weighted configuration model gives the transition, but using the classic, full weighted configuration model always detects modular structure even when none is present (Fig 2a).

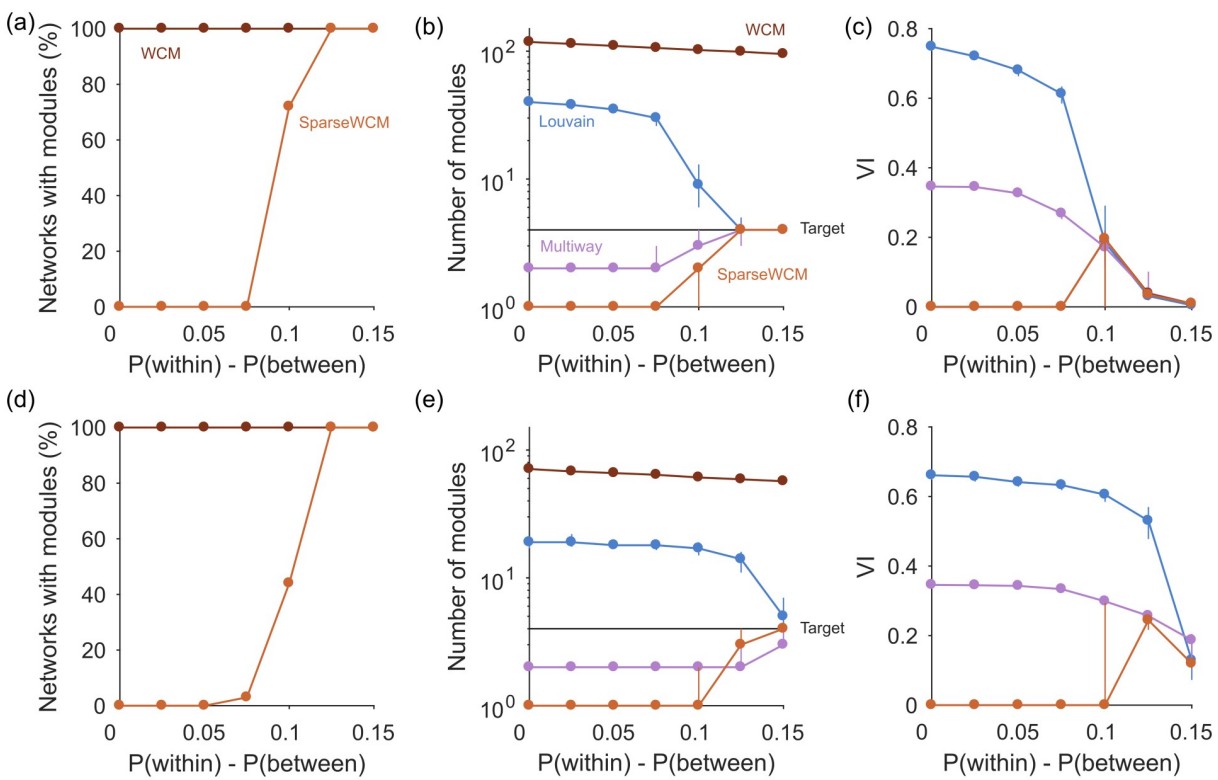

**Fig 2. Performance on synthetic weighted networks.** (a) Proportion of synthetic networks identified as modular by spectral estimation, as a function of the difference in connection probabilities within and between planted modules. All results in (a-c) are from sparse networks with $P$(between) = 0.05, with 100 synthetic networks per $P$(within) tested. (b) Number of modules detected as a function of the difference in connection probabilities. We compare here examples of agglomerative (Louvain) and divisive (multi-way vector) community detection algorithms against the number of communities predicted by spectral estimation. Symbols are medians, bars are inter-quartile ranges over 100 synthetic networks. (c) Performance of community detection, as a function of the difference in connection probabilities. VI: normalised variational information as measure of recovering the ground-truth community assignment [19]; VI = 0 if a partition is identical to the ground-truth (note we score VI = 0 for networks labelled as not modular). The sparse WCM performance is from clustering in the low-dimensional space defined by the null model (see Methods). Symbols are medians, bars are inter-quartile ranges over 100 synthetic networks. (d-f): as (a-c), for denser networks with $P$(between) = 0.15.

When modular structure is detected by the sparse WCM model, the number of eigenvalues $d$ above the null model's estimated upper limit is a good guide to the number of planted communities (Fig 2b). By contrast, using the full configuration model dramatically over-estimates the number of planted communities.

To further illustrate that spectral estimation's ability to distinguish structure is non-trivial, we test examples of standard unsupervised community detection algorithms—Louvain and multi-way spectral clustering—on the same synthetic networks. Both these algorithms always found groups even when the network had no modular structure (Fig 2b). Correspondingly, the accuracy of their community assignment was poor until the synthetic networks were clearly modular (Fig 2c). These results remind us that standard algorithms can give no indication of when a network has no internal structure. We are not claiming spectral estimation to be unique in this regard: other approaches to community detection can also detect transitions between structure and its absence in these simple block models, for example those using the non-backtracking matrix [15] or using significance testing on sampled partitions [20, 21].

When network structure is detected, we have the option of using the $d$-dimensional space defined by spectral estimation to find $d + 1$ groups using simple clustering (see Methods). This approach always performed as well or better than the community detection methods in recovering the planted modules (Fig 2c).

In more densely connected synthetic networks, we find spectral estimation performs similarly in detecting structure, jumping rapidly between rejecting all and accepting all networks as containing modules (Fig 2d). Comparing detection in the sparse (Fig 2a) and dense (Fig 2d) synthetic networks hints that the detectability limit for spectral estimation is constant for the magnitude difference $P(\text{within}) - P(\text{between})$; future work could explore the robustness of this constant limit to changes in the network's parameters, especially size, strength distribution, and number and size of modules. Notably, on these denser networks spectral estimation is always better than community detection alone in detecting both the number of groups (Fig 2e), and in the accuracy of recovering the planted modules (Fig 2f). Spectral estimation can thus successfully both detect low-dimensional structure and its absence at the level of the whole network; we thus next turn to the level of individual nodes.

## Node rejection recovers planted communities among noise

A difficult and rarely tackled problem in analysing networks is the recovery of structure from within noise. Such noise may manifest as extraneous nodes in the network due to sampling only part of the system, or because there really are only a sub-set of nodes contributing to a given structure (e.g. communities). Here we show that our proposed solution of using a low-dimensional space defined by spectral estimation can recover planted network structure from within noise.

We test this by adding a halo of extraneous "noise" nodes to the planted communities in our synthetic networks (Fig 3a). Each synthetic network has $n$ community nodes with planted communities defined by $P(\text{between})$ and $P(\text{within})$, to which we add $n \times f_{\text{noise}}$ additional nodes. The probability of links to, from and between these noise nodes is defined by $P(\text{noise})$. By tuning $P(\text{noise})$ relative to $P(\text{within})$, we can thus move from a strongly modular network when $P(\text{noise}) \ll P(\text{within})$ to a noise-dominated network when $P(\text{noise}) \geq P(\text{within})$.

Fig 3a shows an example such network, with four modules embedded in a set of extraneous nodes, here sparsely connected. We detect these "noise" nodes by projecting all nodes into the $d$-dimensional space defined by the $d$ eigenvalues above the null model's predicted upper limit. As illustrated at the bottom of Fig 3a, nodes not contributing to the low-dimensional structure of the network will cluster close to the origin of this space. We find them by

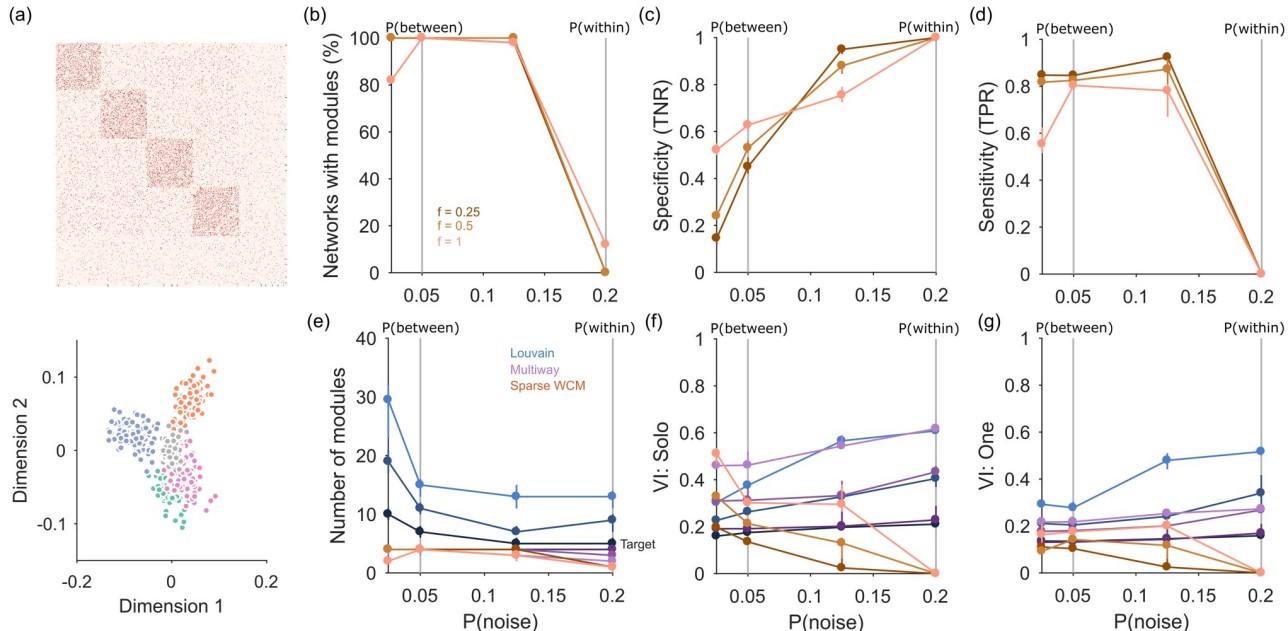

**Fig 3. Node rejection performance.** (a) Top: weight matrix for an example synthetic network with noise, showing the planted modules (block diagonals) and the halo of noise nodes ($P$(noise) = 0.05, $f_{noise}$ = 0.25); throughout we set $P$(between) = 0.05 and $P$(within) = 0.2. Bottom: for the example network, the projection of its nodes onto the first two dimensions retained by spectral estimation (three were found). Nodes are colour-coded by their ground-truth group (colours) or as noise (grey). (b) Proportion of synthetic networks identified as modular by spectral estimation, as a function of the level of noise. Three sizes of noise halo ($f_{noise}$) are plotted. Vertical lines indicate where the link probability for noise nodes was equal to between module ($P$(between)) or within module ($P$(within)) link probabilities. (c) True negative rate (TNR) of node rejection, as a function of the level of noise. TNR is the fraction of correctly rejected noise nodes. Symbols are medians, bars are inter-quartile ranges over 50 synthetic networks. (d) As (c), for the true positive rate (TPR) of node rejection. TPR is the fraction of correctly retained modular nodes. (e) Number of modules detected as a function of the level of noise. For each community-detection algorithm there is one line per $f_{noise}$, with lighter shades for higher $f_{noise}$. Symbols are medians, bars are inter-quartile ranges over 50 synthetic networks. (f) Performance of community detection, as a function of the level of noise. Ground-truth here is with each noise node in its own group. VI: normalised variational information. For each community-detection algorithm there is one line per $f_{noise}$: at a given $P$(noise), higher $f_{noise}$ corresponds to higher VI. Symbols are medians, bars are inter-quartile ranges over 50 synthetic networks. (g) As (f), for ground-truth with a single additional group containing all noise nodes.

predicting the projection of each node from the set of sampled null model networks, and retaining only those nodes whose data projection exceeds the prediction (see Methods). We term this network of retained nodes the "signal" network.

In practice, the combination of spectral estimation and node rejection works well on noisy synthetic networks. The spectral estimation algorithm consistently detects the embedded modular structure when $P$(noise) < $P$(within), and correctly detects the absence of embedded structure when $P$(noise) = $P$(within) (Fig 3b; panel e plots the number of detected modules).

When the embedded network structure is detected, node rejection does well at detecting the noise nodes (Fig 3c), always detecting some noise nodes and thus performing better than without this step. Maximum accuracy at rejection appears to occur at intermediate probabilities of links to and within noise nodes. At the same time, the rejection procedure does well at not rejecting nodes within the embedded modules (Fig 3d).

Again, we can further illustrate the utility of node rejection by looking at the performance of standard community detection algorithms on these synthetic networks with noise. The Louvain algorithm almost always finds too many modules, and both Louvain and multi-way spectral clustering find modules when none exist at $P$(noise) = $P$(within) (Fig 3e). By contrast,

spectral estimation almost always detects the correct number of modules when they are clearly distinguished from the noise nodes (i.e. $P(\text{noise}) < P(\text{within})$, Fig 3e).

To assess the accuracy of community detection, we measure performance against two alternative ground-truths: one where each noise node is placed in its own group; and one where all noise nodes are placed in a single, fifth group. We again also test our simple clustering in the $d$-dimensional space, using the retained nodes; we thus compare to a ground-truth of just the retained nodes. For either ground-truth, the Louvain algorithm performs poorly, and increasingly so as the fraction of noise nodes is increased (Fig 3f and 3g). Multi-way spectral clustering performance is similar to our simple clustering for sparsely connected noise nodes (low $P$ (noise)); with more densely-connected noise nodes, simple clustering in the $d$-dimensional space outperforms the other algorithms at all sizes of the embedding noise (Fig 3f and 3g). The combination of spectral estimation and node rejection thus allows the extraction of embedded community structure in networks.

### Detecting low-dimensional structure in real networks

We now turn to examining what the spectral estimation approach can tell us about real networks, and how our choice of null model affects the conclusions we can draw. To this end, we apply spectral estimation and node rejection to a set of 14 real networks (Table 1), covering all cases of possible weight values (binary, integer, and real-valued).

In Fig 4 we show the null model eigenspectra for this set of networks: the distributions of the eigenvalues of $\mathbf{C}^*$ predicted by the sparse WCM model. Most have a symmetric, narrow-peaked, and heavy-tailed distribution; three are more broadly distributed (Fig 4a). The variation of distribution shape shows the usefulness of the explicit generative approach to estimating the distribution. The distribution of predicted maximum eigenvalues is also approximately symmetric about its mean for most networks (Fig 4b). Setting the upper bound for the real network's eigenvalues as the mean of this distribution is thus a reasonable first approximation.

Setting this upper bound has a dramatic effect on the estimated dimensionality of the real network. One may wonder if it's worth the extra computational effort of estimating the

**Table 1. Real networks and their properties.**

| Name | Size | Links | Density | Link weight |
|---|---|---|---|---|
| Dolphins | 62 | 318 | 0.084 | binary |
| Adjective-Noun | 112 | 850 | 0.068 | binary |
| Power grid | 4941 | 13188 | 0.00054 | binary |
| Star Wars Ep1 | 38 | 270 | 0.19 | integer |
| Star Wars Ep2 | 33 | 202 | 0.19 | integer |
| Star Wars Ep3 | 24 | 130 | 0.24 | integer |
| Star Wars Ep4 | 21 | 120 | 0.29 | integer |
| Star Wars Ep5 | 21 | 110 | 0.26 | integer |
| Star Wars Ep6 | 20 | 110 | 0.29 | integer |
| Les Miserables | 77 | 508 | 0.087 | integer |
| C Elegans[†] | 297 | 4296 | 0.049 | integer |
| COSYNE abstracts | 4063 | 23464 | 0.0014 | integer |
| Political blogs[†] | 1222 | 33428 | 0.022404 | integer |
| Mouse brain gene expression | 625 | $3.9 \times 10^5$ | 1 | real |

All networks were undirected:

[†] indicates a converted directed network by $\mathbf{W} = (\mathbf{W} + \mathbf{W}^T)/2$.

As this conversion can create weights in steps of 0.5, so we used $\kappa = 2$ for these networks.

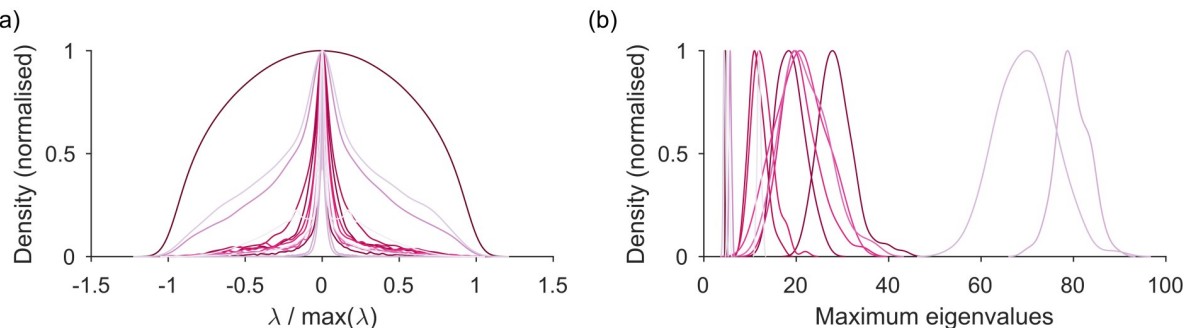

**Fig 4. Distributions of expected eigenvalues for the comparison matrix C of real networks under the sparse WCM null model.** (a) Density of null model eigenvalues $\lambda$ for each network, pooled over all 100 sampled null model networks (here the sparse WCM). Each curve is a kernel density estimate normalised to its maximum. (b) Distribution of the null model's maximum eigenvalue $\lambda^*_{max}$ for each network, over all 100 sampled null model networks.

eigenvalues bounds predicted by the null model. Spectral algorithms for community detection typically use the eigenspectra of $\mathbf{C} = \mathbf{W} - \langle \mathbf{P} \rangle$, using only the expectation $\langle \mathbf{P} \rangle$ of the null model; consequently, all positive eigenvalues of $\mathbf{C}$ are possible dimensions in the network [4, 14]. Fig 5a shows this to be a poor assumption: establishing an upper bound on the expected eigenvalue distribution reduces the estimated dimensionality of the real network (and hence the estimated number of communities) by orders of magnitude (Fig 5a).

The choice of null model to establish the upper bound can strikingly change our conclusions about a given real network. We find the full and sparse WCM models disagree strongly about the dimensionality of some real networks (Fig 5b): notably there are two networks in this data-set where the full WCM model finds more than 35 dimensions, and the sparse WCM model finds at most one. The sparse WCM model mostly estimates fewer dimensions, consistent with its closer estimates of the real network's sparseness, and its more accurate performance on synthetic data (Fig 2). Most striking is that we are able to reject the existence of low-dimensional community structure for a handful of the real networks (0's in Fig 5b), but the null models do not agree on which networks have no structure (no real network has 0's for both null models in Fig 5b). These results underline how the choice of null model is critical when testing the structure of a network.

**Node rejection stabilises analysis of real networks.** When we then test for node rejection using the sparse WCM model, all real networks with low-dimensional structure have nodes rejected. The resulting signal network is up to an order of magnitude smaller than the original network (Fig 5c).

This offers some straightforward but nonetheless useful advantages. As we demonstrate below, one advantage is that the signal network can simplify interpretation of the network's structure. Another advantage is that it reduces the variability of unsupervised analyses of the network. To demonstrate this, we apply the Louvain algorithm to the full and signal versions of each real network. As expected, the number of modules detected in the signal networks is usually—but not always—smaller than in the full network (Fig 5d). Over repeated runs of the Louvain algorithm, the range of detected modules can vary considerably in the full real networks, but this variation is markedly reduced for the signal versions of the same network (Fig 5e).

**Hidden *k*-partite structure in real networks.** Throughout this paper we estimate the upper bound of the eigenvalue spectrum predicted by the null model; but we could equally well estimate the lower bound, and check if the data network at hand has eigenvalues that fall

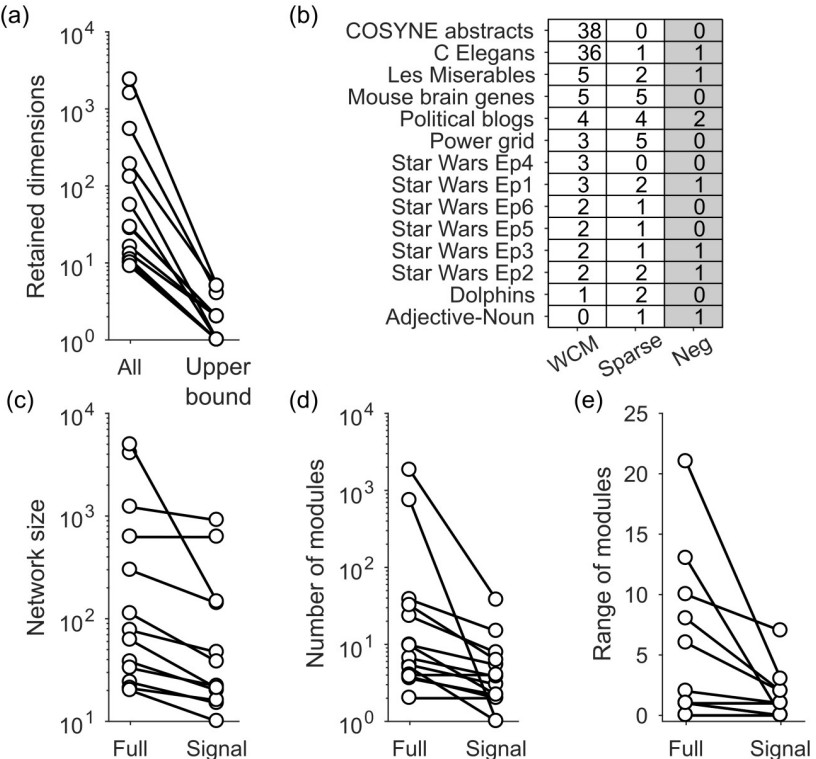

**Fig 5. Detecting low-dimensional structure in real networks using spectral estimation.** (a) Number of retained dimensions for each network when using spectral estimation to obtain an upper bound on the null model, against using all positive eigenvalues (for the sparse WCM). Note some networks have no retained dimensions when using spectral estimation, so appear only in the "All" column. (b) Number of retained dimensions for each network when using the full ('WCM') or sparse weighted configuration model; the third column ('Neg') gives the number of retained dimensions below the predicted lower bound of the eigenvalue spectrum, for the sparse WCM. (c) Number of nodes in each full network against the number of nodes in the signal network (those remaining after node rejection in the low-dimensional space). Data for the sparse WCM. (d) Mean number of modules found in the full or signal version of each network (Louvain algorithm). (e) Range of the number of modules found across five runs of the Louvain algorithm, in the full and signal versions of each network.

below this lower bound. Real networks with eigenvalues of **C** below the lower bound indicate $k$-partite structure [4]. In the simplest case, one eigenvalue below the lower bound is evidence of bipartite structure ($k = 2$), with two groups of nodes that have more connections between the groups and fewer within each group than predicted by the null model.

When we use the sparse WCM to estimate the predicted lower bound of the real networks here, we find seven have eigenvalues below that lower bound. All but one of those networks have just one eigenvalue, and so are bipartite (third column in Fig 5b). Applying node rejection to the corresponding eigenvector rejects a considerable proportion of nodes, indicating that the $k$-partite structure is embedded within the network; we show examples in Section 2 in S1 Appendix. Thus, spectral estimation using the lower bound can reveal hidden $k$-partite structure in larger networks.

## Insights into specific networks

We now look in more detail at examples from the data-set of real networks to illustrate the new insights brought by spectral estimation (of the upper bound). Any interpretation of global

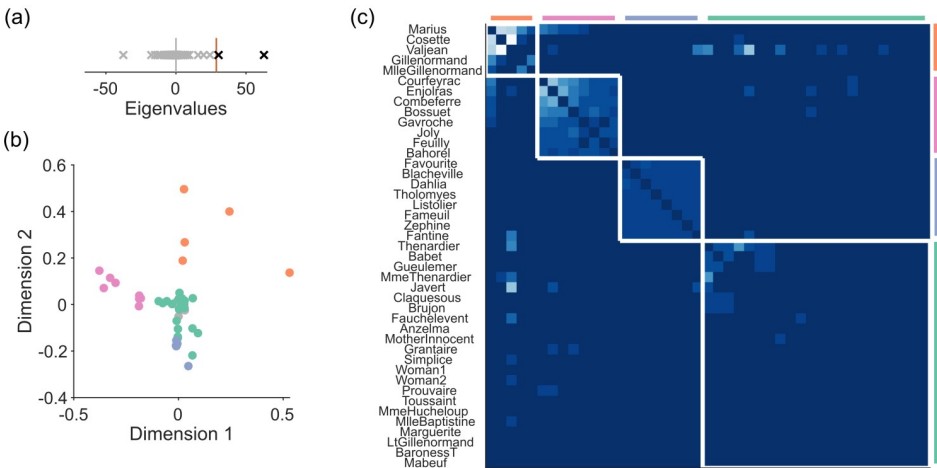

**Fig 6. Spectral estimation of Les Miserables scene network.** (a) Eigenvalues of Les Miserables' comparison matrix, and the maximum eigenvalues predicted by the sparse WCM model (red line). (b) Projection of all nodes into the two dimensional space defined by the retained eigenvectors. Colours correspond to the modules in panel (c). Rejected nodes are coloured grey, and are centred at the origin. (c) Modules found by consensus clustering in the low-dimensional projection. A heatmap of **W** for the signal network, ordered by detected modules (white boxes). Lightness of colour is proportional to the integer weights between nodes.

structure in real networks faces the problem that meta-data about network nodes can be a poor guide to ground-truth [22]—and indeed that there need exist no "ground-truth". Thus here we use domain knowledge to aid interpretation of the results. We first look at networks derived from a narrative structure in order to compare the recovered signal network and its modules to the narrative.

**Les Miserables narrative.** The Les Miserables network encapsulates the book's narrative by assigning characters to nodes and a weighted link between a pair of nodes according to the number of scenes in which that pair of characters appear together. Spectral estimation detects a departure from the sparse WCM model (Fig 6a), and hence a low-dimensional structure to the narrative (Fig 6b). Node rejection in this two dimensional space removes 30 nodes, yet retains all major characters (for example, Valjean, Marius, Fantine, and Javert), considerably simplifying the identification of the main narrative structure.

We use unsupervised consensus clustering on the low-dimensional projection of the signal network in order to identify small modules potentially below the resolution limit. This recovers four modules, corresponding to major narrative groups, including Les Amis de l'ABC (the "Barricade Boys": Enjoiras and company), and the student friends of Fantine (Fig 6c). Thus for the Les Miserables network, spectral estimation can correctly identify the major characters, and identifies key narrative groups.

**Star Wars dialogue structure.** The networks of dialogue structure in Star Wars Episodes 1 to 6 illustrate how we can detect qualitative differences in narratives using spectral estimation. In each of these six networks, each node is a character in that film, and the weight of each link between nodes is the number of scenes in which that pair of characters share dialogue.

Applying spectral estimation to each film's network reveals that only four of the six have a low-dimensional structure beyond that predicted by the sparse WCM model (Fig 7a). Character interactions in Episode 4 (A New Hope) and Episode 6 (Return of the Jedi) do not depart from the null model. From this we might conclude that the complexity of dialogue structure is no predictor of the quality of Star Wars films. Plotting the strength of departure from null

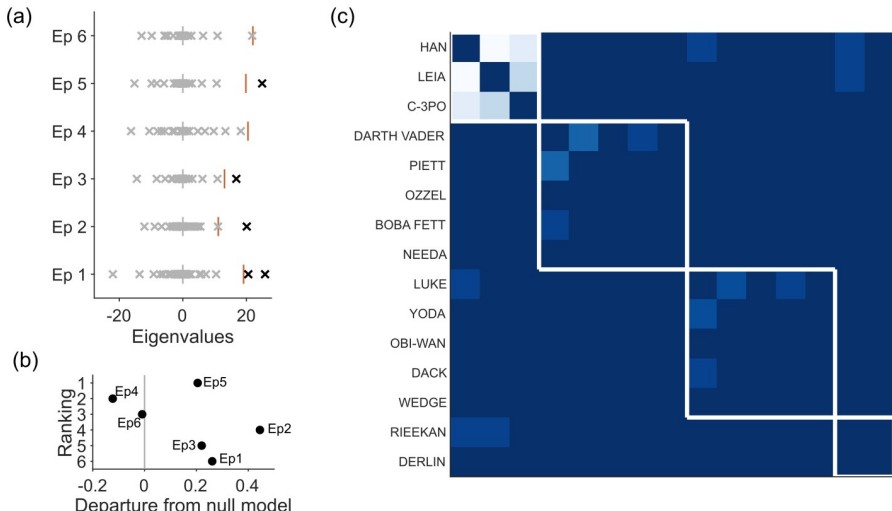

**Fig 7. Star Wars dialogue networks for Episodes 1–6.** (a) Eigenvalues of each episode's comparison matrix, and the maximum eigenvalues predicted by the sparse WCM model (red line). Two of the original trilogy do not exceed the maximum eigenvalue predicted by the null model. (b) Departure from the null model against film ranking (Ranking source: https://www.theguardian.com/film/2018/may/24/every-star-wars-film-ranked-solo-skywalker). Departure is: $[\lambda_{max} - \langle \lambda^*_{max} \rangle]/\lambda_{max}$, the distance between the data's maximum eigenvalue $\lambda_{max}$ and the predicted upper bound from the null model, normalised. (c) Modules in Episode 5 (The Empire Strikes Back), found by consensus clustering of the low-dimensional projection. Each module corresponds to a story arc.

model against a respected critic's ranking of the films' quality supports this conclusion (Fig 7b).

Nonetheless, when there is low-dimensional structure, consensus clustering of the signal network recovers modules that correspond to narrative arcs in each film. In Fig 7c we illustrate this for Episode 5 (The Empire Strikes Back), where the clustering recovers the separate arcs of the fleeing Millennium Falcon, Luke on Dagobah, and the Empire and its associates.

Notably, Star Wars Episodes 1–3 are also the only ones to have a bipartite structure (see Section 2 in S1 Appendix), indicating an overly-structured narrative in which there exists both well-defined groups of characters that converse, and well-defined groups that do not interact at all.

**Co-author network of the COSYNE conference.** Networks of scientific fields are useful surrogates for social networks as we can bring considerable domain knowledge to bear on their interpretation. As an example of this, here we take a look at the network of co-authors at the annual, selective Computational and Systems Neuroscience (COSYNE) conference. This network's nodes are authors of accepted abstracts in the years 2004–2015, and the weight of links between authors is the number of co-authored abstracts in this period. The full network has 4806 nodes, from which we analyse the largest component containing 4063 nodes.

As shown in Fig 5b, using the full configuration model as the null model for spectral estimation predicts 38 dimensions in this network. If we run the Louvain algorithm on the full network, it finds 728 modules. This order-of-magnitude discrepancy in the predicted dimensions and detected modules is reminiscent of the poor performance in estimating modules that we observed in Fig 2b for synthetic networks without modular structure.

Indeed, when we instead use the sparse WCM as the null model for spectral estimation, no low-dimensional structure is found. And this is useful, as it suggests this particular null model captures much of the structure of the real network. Here the sparse WCM model suggests that

the collaborative structure in the COSYNE conference is no different to a model where, once a pair of authors have begun working together, then the number of co-authored abstracts by that pair is simply proportional to their total output. The consequent absence of low-dimensional structure suggests there is no rigid subject-based division (into e.g. vision and audition; or cortex and hippocampus) of this conference network.

**Gene co-expression in the mouse brain.** Our final detailed example demonstrates the use of spectral estimation on a general clustering problem. The Allen Mouse Brain Atlas [23] is a database of the expression of 2654 genes in 625 identified regions of the entire mouse brain. From this database, we construct a network where each node is a brain region, and the weight of each link is the Pearson's correlation coefficient between gene-expression profiles in those two regions. One goal of clustering such gene co-expression data is to detect correspondences between gene expression and brain anatomy [24].

An advantage of using spectral estimation on such a clustering problem is the unsupervised detection of the dimensionality of any clusters. Using sparse WCM as the null model, we find the gene co-expression network has five eigenvalues above the expected upper limit (Fig 8a). Projection of the nodes onto the first two of the five retained dimensions indicates a clear group structure (Fig 8b). Reassuringly, no nodes are rejected from this network. (For if we found rejected nodes here, it would mean either that small brain regions existed with profiles of gene expression that bore no resemblance to others, which would be difficult to reconcile with known patterns of brain development; or that there was a considerable error in those regions' gene expression profiling).

Fig 8c plots the partition with maximum modularity that we found by clustering in this five-dimensional space (consensus clustering gives us 26 groups, which are subdivisions of

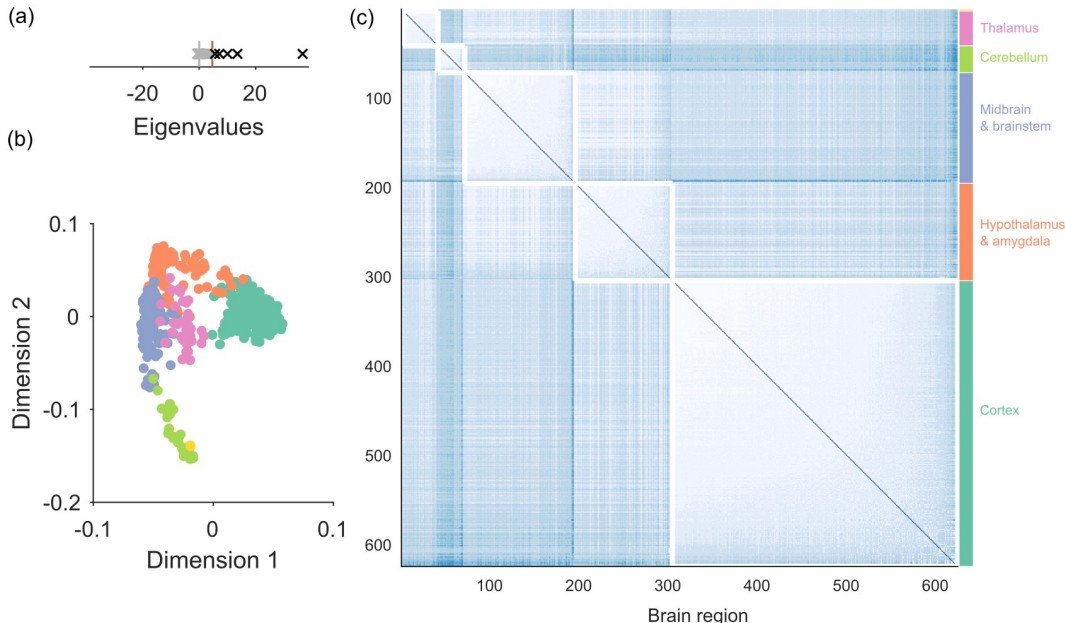

**Fig 8. Network of gene expression in the mouse brain.** (a) Eigenvalues of the gene expression network's comparison matrix, and the maximum eigenvalues predicted by the sparse WCM model (red line). (b) Projection of all nodes into the two dimensional space defined by the top two retained eigenvectors. Colours correspond to the modules in panel (c). (c) Modules found by clustering in the complete five-dimensional projection. The detected modules map the 625 brain regions onto highly distinct regions of neural tissue.

these groups). As shown on the figure, the detected modules correspond remarkably well to highly distinct broad divisions of the mammalian brain.

## Discussion

Detecting low-dimensional structure in a network requires a null model for the absence of structure. The choice of null model in turn will define the type of structure that can be detected. Here we introduced a spectral approach to detecting low-dimensional structure using a chosen generative null model. We have shown that this spectral approach allows rejection and detection of structure at the level of the whole network and of individual nodes.

Our results emphasised that the choice of null model can strongly change conclusions about network structure. Here we contrasted the classic configuration model with a sparse variant that accounted for the problem that using the classic configuration model as a generative model creates networks that are denser than the data network at hand. Indeed, for the synthetic networks, using the classic configuration model consistently predicts a vastly more complex structure than actually exists. By contrast, using the sparse variant correctly detects the absence of structure in synthetic networks, and sharply transitions to detecting community structure when present. It also reveals the absence of community structure in a set of real networks. Using analysis of network structure to do scientific inference will thus need careful choice of an appropriate null model.

Indeed there are now a wide range of null model networks to choose from. Variations of the configuration model abound [10, 12], including versions for correlation matrices [25], and simplical models [26]. Other options include permutation null models, derived directly from data networks by the permutation of links [27], and specific generative models for network neuroscience applications [28]. Exploring the insights of these null models in our spectral estimation approach could be a fruitful path.

We illustrated the advantages of using spectral estimation over naive community detection, using the Louvain algorithm and multi-way spectral clustering as examples of unsupervised agglomerative and divisive approaches. In particular, we demonstrated that "noise" nodes without community assignment are dealt with by spectral estimation, but invisible to standard algorithms, which can result in them performing poorly in finding the number and membership of true communities. This raises questions over the accuracy of standard algorithms' results on real network data, in which the prevalence of "noise" nodes is rarely assessed. Indeed, our analysis of real-world data suggests all but one data network we tested contains a substantial fraction of noise nodes—and the one that did not was a network of gene expression correlations across brain regions, for which finding "noise" nodes would have indicated an error in our approach.

Once we have derived the signal network using spectral estimation, we can apply any unsupervised community detection algorithm to it, including the Louvain and multi-way spectral clustering. And indeed for multi-way spectral clustering, we can specify the number of groups to find directly from the number of dimensions found by the spectral estimation algorithm. Of course, this does not change the limitation that any community detection algorithm that maximises modularity contends with the twin problems of the resolution limit [29], and the degeneracy of high values for modularity [30]. For our analyses of real networks, we supplemented our community detection with consensus clustering to address these issues.

Our work here continues a considerable body of work using spectral approaches to detecting the number of communities in a network [4, 13, 31, 32]. A recent breakthrough has been the idea of non-backtracking walks on a network, as the eigenvalues of the corresponding matrix can detect community structure in synthetic sparse networks down to the theoretical

limit [15–17]. Our work complements this prior work by allowing a choice of null models to define structure at the network level, and goes beyond them by creating an approach for rejecting nodes.

While we have focussed here on community detection, in principle the type of structure we can detect depends on the choice of null model. For example, we could fit a stochastic block model to our data network [33–35], and use this fitted model to generate our sample null model networks $\mathbf{P}^*$. The spectral estimation algorithm would then test the extent of the departure between the data network and the fitted block model. Similarly, one could use fitted core-periphery models [36] as the generative null model, and test departures from this structure in the data.

While our approach allows the use of any null model, this also brings the inherent limitation that we must explicitly generate a sample of null model networks. The computational cost scales with both the number and size of the generated networks. However, as we discuss in detail in the Methods, there are a number of approaches to ameliorate the computation time; indeed, the primary bottleneck in our code implementation is not computation time but memory for storage of all generated networks (see Methods section "Practical computation of the sampled null models" for details). For scaling our approach to networks of $n \gg 10^4$, future development of more efficient memory usage is a priority.

Of the other possible developments of our spectral estimation approach, two stand out. One is that we could construct a de-noised comparison matrix from the outer product of the eigenvectors retained by spectral estimation. This approach has been successfully applied to analyses of both financial [37, 38] and neural activity [39, 40] time-series, where de-noised matrices of time-series correlation allow for more accurate inference of structure. Another is to further develop node rejection. Here we tested an initial idea of comparing projections between the data and null models, which performed reasonably well, with clear scope for improved performance with more rigorous approaches. These and other potential developments suggest that our spectral estimation approach is a promising basis for richer comparisons of real-world networks with suitable generative models.

## Methods

We develop our spectral algorithm for weighted, undirected networks. For a given network of $n$ nodes, we will make use of both its adjacency matrix $\mathbf{A}$, whose entry $A_{ij} \in \{0, 1\}$ defines the existence or absence of links between nodes $i$ and $j$, and the corresponding weight matrix $\mathbf{W}$, whose entry $W_{ij}$ defines the weight of the link between nodes $i$ and $j$. Weights can real valued or integer valued; in the latter case the weights are equivalent to the number of links between $i$ and $j$. For binary networks $\mathbf{A} = \mathbf{W}$.

Detecting the existence of low-dimensional structure in networks requires that we compare the data network with some null model for the absence of that low-dimensional structure. A simple comparison is

$$\mathbf{C} = \mathbf{W} - \langle \mathbf{P} \rangle, \tag{1}$$

where $\langle \mathbf{P} \rangle$ is an expectation of the link weights over the ensemble of possible networks consistent with the chosen null model. The comparison matrix $\mathbf{C}$ thus encodes the departure of the data network from the null model. If we choose the classic configuration model as the null model, then $\mathbf{C}$ is the well-known modularity matrix [4]. But we'd like the freedom to choose the most appropriate null model, according to the structural hypothesis we want to test.

In seeking to detect low-dimensional structure, it is particularly useful that the eigenvalue spectrum of $\mathbf{C}$ contains much information about the structure of the network [4]. In general,

the separation of a few eigenvalues from the bulk of the spectrum indicates low-dimensional structure in a matrix [37–40]; for networks, this can indicate the number of communities within it, and form the basis of a low dimensional projection of the network [4, 13, 14]. So our goal is to estimate the spectrum of **C** predicted by a given class of null model, and compare it to the spectrum of **C** for the data network; a departure between the predicted and data spectra then indicates the presence of meaningful structure in the network. It also gives us additional information about the data network, as we detail below.

Our general approach then is to sample null model networks from a generative model with expectation $\langle \mathbf{P} \rangle$, and so sample the expected variation in **C** solely due to the ensemble of networks consistent with the null model. We then use these samples to estimate the eigenspectrum of **C** due solely to variations in the networks consistent with the null model. In particular, we estimate its upper bound: if any of the data network's eigenvalues exceed that bound, we have evidence of low-dimensional structure in the data network **W** that is not captured by the ensemble of null model networks. Moreover, data eigenvalues that exceed the limits predicted by the model provide us with additional information about the structure of the data network; and, as we show below, a basis for testing node-level membership of a network too.

### The spectral estimation algorithm

Our spectral estimation algorithm proceeds as follows. Given some chosen generative null model, we:

1. generate $N$ sample null model networks $\{\mathbf{P}_1^*, \mathbf{P}_2^*, \ldots, \mathbf{P}_N^*\}$.

2. from each we can then compute the sampled network's comparison matrix $\mathbf{C}_i^* = \mathbf{P}_i^* - \langle \mathbf{P} \rangle$, for $i \in \{1, 2, \ldots, N\}$,

3. and the comparison matrices' corresponding set of eigenvalues $\{\lambda_1^*, \lambda_2^*, \ldots, \lambda_n^*\}_i$, for the $i$th sampled network.

4. These sets of eigenvalues $\{\lambda_1^*, \lambda_2^*, \ldots, \lambda_n^*\}_1 \ldots \{\lambda_1^*, \lambda_2^*, \ldots, \lambda_n^*\}_N$ are then samples from the eigenspectrum of the population of null model networks whose expectation is $\langle \mathbf{P} \rangle$. In practice here we make use of the upper and lower bounds, and so estimate them directly. We denote the maximum eigenvalue from each of the $N$ sampled networks as $\lambda_{\max}^*(i)$. The upper bound of the eigenspectrum predicted by the null model is estimated as the expectation $\langle \lambda_{\max}^* \rangle$ over those $N$ maximum eigenvalues. Similarly, the lower bound is estimated as $\langle \lambda_{\min}^* \rangle$ over the set of $N$ minimum eigenvalues.

For comparison, we compute the data's comparison matrix $\mathbf{C} = \mathbf{W} - \langle \mathbf{P} \rangle$, and its eigenvalues $\lambda_1, \lambda_2, \ldots, \lambda_n$. How we compute $\langle \mathbf{P} \rangle$ will depend on the null model at hand: for some, $\langle \mathbf{P} \rangle$ is known analytically; for others we can estimate it as the expectation over $\{\mathbf{P}_1^*, \mathbf{P}_2^*, \ldots, \mathbf{P}_N^*\}$.

With these to hand, we then test our null model (Fig 1a). If any data eigenvalues exceed the expected upper bound $\langle \lambda_{\max}^* \rangle$, then we have evidence that the data network contains low-dimensional structure not captured by the null model. If not, then we cannot discount that the data network is a realisation of the null model **P**.

(Note that we treat this process here as an estimation problem: in Section 3 in S1 Appendix, we outline how the same process can be cast in a null hypothesis significance testing framework, to test the rejection of each data eigenvalue at some defined $P$-value).

For a data network that departs from the null model, we will have $d$ eigenvalues of the data network greater than $\langle \lambda_{\max}^* \rangle$, labelled $\lambda_1, \lambda_2, \ldots, \lambda_d$, which estimate the dimensionality of the data network with respect to the null model. The corresponding eigenvectors $\mathbf{u_1}, \mathbf{u_2}, \ldots, \mathbf{u_d}$

define a $d$-dimensional space into which we can project the data network's comparison matrix $\mathbf{C}$ (Fig 1b): we can then use this projection to infer properties of the structure of $\mathbf{W}$ (detailed below for community detection), and perform a rejection test per node.

**Node rejection.** Each node will have a projection in this $d$-dimensional space. Nodes that are weakly contributing to the structure of the network captured by this space (e.g. nodes that are not in any community) will have small values in each eigenvector, and so have short projections that remain close to the origin (Fig 1c). We can thus reject individual nodes by defining a boundary on "close".

Here we do this by comparing the data network's projections to those predicted by the sampled null model networks. For node $j$, we compute its L2 norm from the $d$-dimensional projection of $\mathbf{C}$: $L(j) = \sqrt{\sum_{i=1}^{d}[\lambda_i \mathbf{u_i}(j)]}$. We also compute the L2 norm for the $j$th node in the $d$-dimensional projection of each $\mathbf{C}^*$ obtained from the $N$ sampled networks, giving the distribution $L(j)_1^*, L(j)_2^*, \ldots, L(j)_N^*$ over all sampled networks. From that distribution, we compute the expected projection $\langle L(j)^* \rangle$ for that node. A node is then rejected if $L(j) < \langle L(j)^* \rangle$, which defines here our boundary on "close" to the origin; otherwise the node is retained. Again we frame this here as an estimation problem; interesting extensions of this work would be to test node rejections based on confidence intervals or hypothesis tests that make use of the null model distribution of projections $L(j)_1^*, L(j)_2^*, \ldots, L(j)_N^*$.

We call the retained nodes the "signal" network. Rejecting nodes from a sparse network may fragment it; it may also leave isolated leaf nodes with a single link to the rest of the network. Consequently, in practice, we strip the leaf nodes and retain the remaining largest component as the "signal" network.

## Generative null models

Key to our algorithm is the use of a generative null model for sampling networks. We use two generative models here, based on the classic configuration model.

**Weighted configuration model.** We start with the weighted version of the classic configuration model [10, 14, 18]. In this model, the strength sequence of the network is preserved, and the expectation $\langle \mathbf{P} \rangle$ is $P_{ij} = s_i s_j / w$, where $s_i, s_j$ are the strength of nodes $i$ and $j$, and $w$ is the sum total of unique weights in the network. (We also tested the computation of $\langle \mathbf{P} \rangle$ as the expectation over the $N$ sampled networks, for consistency with how we computed it for the sparse model below; we obtained identical results).

**Sparse weighted configuration model.** The classic weighted configuration model is dense, as the expectation $\langle \mathbf{P} \rangle$ has an entry for every pair of nodes. However, real networks are predominantly sparse [1, 2]. Consequently each sampled network using the weighted configuration model is also likely more densely connected than its corresponding data network. This difference is amplified in weighted networks because the comparatively denser connections in the sample network means the weights are spread over more links than in the data network, creating a potentially large difference in the distribution of weights. We show this large disagreement for an example real network in Fig 1 in S1 Appendix.

To better take into account the distribution of link weights and sparseness, we use a sparse weighted configuration model (sparse WCM). This model generates a sample network in two steps. We first create the sampled adjacency matrix $\mathbf{A}^*$ using the probability of connecting two nodes $p(\text{link}|i, j) = k_i k_j / 2m$, where $k_i$ is the degree of node $i$, and $m$ is the total number of unique links in the data adjacency matrix $\mathbf{A}$. We then create the sampled sparse weight matrix $\mathbf{P}^*$ by assigning weights only to links that exist in $\mathbf{A}^*$, as detailed below. This is repeated $N$

times. In the absence of an analytical form for $\langle \mathbf{P} \rangle$, we compute it as the expectation over the $N$ generated networks, with elements $\langle P_{ij} \rangle = \frac{1}{N} \sum_{k=1}^{N} P_{ij}^*(k)$.

**Poisson generation of links.** When weights between nodes are binary or integer valued, then an exact way of generating a sample network $P^*$ from these null models is by stub-matching, where node $i$ is assigned $s_i$ stubs, and stubs are linked between nodes at random until all stubs are matched. Stub-matching in the sparse model would be restricted to the linked nodes in the sampled adjacency matrix $\mathbf{A}^*$. While we provide code for building these models using stub-matching, generative procedures using stub matching can be prohibitively slow with many links, many nodes, or real-valued weights converted to integers—all of which we have here.

We thus use a Poisson model for drawing the link weight between any pair of nodes (the use of the Poisson model is motivated by the fact that all networks we deal with have, or are converted to, integer weights—see below). We draw the weight between nodes $i$ and $j$ from a Poisson distribution with

$$\lambda_{ij} = N_{\text{link}} p(\text{link}|i, j), \tag{2}$$

where $p(\text{link}|i, j)$ is the probability of placing a single link between nodes $i$ and $j$, and $N_{\text{link}}$ is the total number of links to place in the null model network $P^*$.

For both classic and sparse weighted configuration models, it follows from the stub-matching model that the probability of placing a link between a pair of nodes is:

$$p(\text{link}|i, j) = \frac{s_i s_j}{\sum_{ij \in \mathbf{A}^*} s_i s_j}. \tag{3}$$

In the classic configuration model all pairs of nodes are linked in $\mathbf{A}^*$, so the sum in the denominator of Eq 3 is over all pairs of nodes; the total number of links to place is then $N_{\text{link}} = \frac{1}{2} \sum_{i}^{n} s_i$.

In the sparse model, only a subset of nodes in $\mathbf{A}^*$ are linked, so the sum in the denominator of Eq 3 is over just those linked nodes. We then generate $\mathbf{P}^*$ by drawing weights only for those pairs of linked nodes, with the total number of links to place being $N_{\text{link}} = \frac{1}{2} \sum_{i}^{n} s_i - 2m$, where $m$ is the number of unique links already in $\mathbf{A}^*$.

As well as dramatically speeding up computation time, this Poisson approach has two appealing features. First, it gives a model that is closely linked to the generative process of many real-world networks, for which weights are counts of events in time or space (e.g. word co-occurrence; co-authorships; character dialogue). Second, it also closely approximates the multinomial distribution of link weights that results from stub-matching ($M(N_{\text{link}}, \{p(\text{link})_1, p(\text{link})_2, \ldots, p(\text{link})_m\})$, for all $m$ unique links), becoming arbitrarily close as $N_{\text{link}} \to \infty$.

**Practical computation of the sampled null models.** The Poisson model and stub-matching work for binary or integer weights. To deal with data networks of real-valued weights, we first quantise them to integer values: we scale all weights by a conversion factor $\kappa > 1$ and round to get an integer weight. Once all links are placed, we then convert back to real-valued weights by rescaling all weights by $1/\kappa$. The choice of $\kappa$ is strongly determined by the discretisation and distribution of weights. For networks with weights in steps of 0.5, we use $\kappa = 2$; for networks based on similarity $\in [0, 1]$ we use $\kappa = 100$ (which implies that weights less than 1/100 are not considered links). These scalings are used for all types of generative model in this paper.

Typically we generate $N = 100$ null models for each comparison with a data network. The generative model approach is of course more computationally expensive than using just the expectation of the null model $\langle \mathbf{P} \rangle$ in Eq 1. However, as each generated null model network is

an independent draw from the ensemble of possible networks, this process is easily parallelised; all code was run on a 12-core Xeon processor. Moreover, the Poisson model is quick; even our largest weighted network (4096 nodes) took a few seconds to generate each null model network.

Rather, a potential bottleneck for scaling our spectral estimation algorithms is memory (RAM). For example, given a data network of $n$ nodes we create a $n \times n \times N$ matrix of sampled weight matrices (and the same size matrix of eigenvectors). For a data network of $n = 10^5$ nodes and $N = 100$ sampled null models, the sampled weight matrices would require 74.5 GB at the default double-precision float used in our code; an immediate improvement in memory could be had if the data networks used binary or integer weights, and so could be cast to the appropriate data class: using unsigned 8-bit integers for a binary network of that size would require 9.3GB; unsigned 16-bit integers for an integer-weighted network of that size would require 18.6GB. But even then, a data network of $n = 10^6$ nodes, which are common, would require 930GB just to store the 100 generated null models for a binary network. Further improvements to the algorithm's code implementation could also create more efficient memory usage by, for example, first taking a two step approach of generating only the eigenvalues to do spectral estimation, then generating only the specified number of leading eigenvectors.

## Testing the spectral estimation algorithm's performance

**Synthetic networks.** We use a version of the weighted stochastic block model to test our spectral estimation algorithm. We specify $g$ modules of size $\{N_1, \ldots, N_g\}$. Here each synthetic network has $n = 400$ nodes divided into $g = 4$ equal-sized groups. Its adjacency matrix $\mathbf{A}_{\text{sbm}}$ is constructed by creating links between groups with probability $P(\text{between})$ and within groups with probability $P(\text{within})$. The weight matrix $\mathbf{W}_{\text{sbm}}$ is then constructed by first sampling a strength sequence $s_1, \ldots, s_n$ from a Poisson distribution with parameter $\lambda_s$ ($\lambda_s = 200$ throughout). We then sample weights from a Poisson distribution: for each link $(i, j)$ in $\mathbf{A}_{\text{sbm}}$, we draw a weight from the Poisson distribution $\lambda = N_{\text{link}} \, p(\text{link}|i, j)$, exactly as for the sparse WCM. Note we deliberately construct the synthetic networks as sparse weighted networks in order to detect any differences in performance between the null models.

To test rejection of nodes not contributing to a network's low-dimensional structure, we add a noise halo to our stochastic block model. We add $n \times f_{\text{noise}}$ noise nodes to the synthetic network, to give $T = n + \lfloor n \times f_{\text{noise}} \rfloor$ nodes in total. To construct $\mathbf{A}_{\text{sbm}}$, the first $n$ nodes have the above modular structure defined by $P(\text{within})$ and $P(\text{between})$; the additional noise nodes are connected to all other nodes, including each other, with probability $P(\text{noise})$. The weight matrix $\mathbf{W}_{\text{sbm}}$ is then constructed as above, sampling the strength sequence of all $T$ nodes from a Poisson distribution, and the consequent weights conditioned on the links in $\mathbf{A}_{\text{sbm}}$. Thus, both modular and noise nodes have the same expected strengths, differing only in the distribution of their weights.

**Community detection algorithms.** As benchmarks for community detection performance we use the standard Louvain algorithm [41] as an example of an agglomerative algorithm, and multi-way vector-partition [42] as an example of a divisive algorithm. We introduce an unsupervised version of this multi-way vector algorithm in Section 4 in S1 Appendix.

As our spectral estimation procedure will be estimating the exact number of communities $c$, we also want a way to do community detection given the $d$-dimensional projection of $\mathbf{C}$. We use a simple clustering in this space [14]. We project all nodes using the $d$ eigenvectors, and k-means cluster $p = 100$ times, given $c$ clusters as the target and using Euclidean distance between the nodes. For each partition, node assignment to the $c$ communities is encoded in

the binary matrix $\mathbf{S}$ with $S_{ij} = 1$ if node $i$ is in community $j$, and $S_{ij} = 0$ otherwise [4]; from this we compute the modularity $Q$ of each partition as $Q = \text{Tr}(\mathbf{S}^{\mathbf{T}}[\mathbf{W} - \langle \mathbf{P} \rangle]\mathbf{S})$, where Tr is the trace operator, and using the expectation $\langle \mathbf{P} \rangle$ over our chosen null model. We retain the partition that maximises $Q$.

For real networks, we address the resolution limit [29] and degeneracy of maximal $Q$ solutions [30] by also using our unsupervised consensus clustering approach [43], which we extend here to use an explicit null model for consensus matrices. Briefly, given the $p$ partitions, we construct a consensus matrix $\mathbf{D}$ whose entry $D_{ij} = n_{ij}/p$ is the proportion of times nodes $i$ and $j$ are in the same cluster. We construct the consensus comparison matrix $\mathbf{C}_{\text{con}} = \mathbf{D} - \mathbf{P}_{\text{con}}$, given a specific null model for consensus clustering (defined below). As the purpose of using the consensus clustering is to explore more and smaller module sizes than can be accessed by maximising $Q$ alone, we use the number of positive eigenvalues $K$ of $\mathbf{C}^{\text{con}}$ as the upper limit on the number of modules to check. That is, we project $\mathbf{C}^{\text{con}}$ using the $K$ top eigenvectors, then use k-means to cluster the projection of $\mathbf{C}^{\text{con}}$ $p = 100$ times for each $k$ between 2 and $K$. From these $p$ $(K - 1)$ partitions, we construct a new consensus matrix. The general consensus null model is the proportion of expected co-clusterings of a pair of objects in the absence of cluster structure, with entries: $P_{ij}^{\text{con}} = 1/(p(K - 1)) \sum_{c=l}^{K} p/c$, where the sum is taken over all tested numbers of clusters $c$ from some lower bound $l$ ($l = K$ for the initial consensus matrix above; $l = 2$ otherwise). We repeat the consensus matrix and clustering steps until $\mathbf{C}_{\text{con}}$ has converged on a single partition.

**Data networks.** All real-world networks were checked for a single component: if not connected, then we used the giant component as $\mathbf{W}$ for spectral estimation.

The following networks were obtained from Mark Newman's repository (http://www-personal.umich.edu/~mejn/netdata/): the Les Miserables character co-appearances; the dolphin social network of Doubtful Sound, New Zealand [44]; the adjective-noun co-occurrence network of David Copperfield; the USA 2004 election political blogs network; the C Elegans neuronal network, and the Western USA power grid [45].

Networks of shared character dialogues in Star Wars Episodes I-VI were constructed by Evelina Gabasova [46], and are available at http://doi.org/10.5281/zenodo.1411479.

Data on abstract co-authorship at the annual Computational and Systems Neuroscience (COSYNE) conference were shared with us by Adam Calhoun (personal communication). These data contained all co-authors of abstracts in each of the years 2005 to 2014. From these we constructed a single network, with nodes as authors, and weights between nodes indicating the number of co-authored abstracts in that period.

We obtained the Mouse Brain Atlas of gene co-expression [23] from the Allen Institute for Brain Sciences website (http://mouse.brain-map.org/), using their API. The Brain Atlas is the expression of 2654 genes in 1299 labelled brain regions. However, these regions are arranged in a hierarchy; we used the 625 individual brain regions at the bottom of the hierarchy as the finest granularity contained in the Atlas. We constructed the gene co-expression network by calculating Pearson's correlation coefficient between the gene expression vectors for all pairs of these 625 brain regions; all correlations were positive.

## Resources

MATLAB code implementing the spectral estimation algorithms, synthetic network generation, and scripts for this paper are available under a MIT License at https://github.com/mdhumphries/NetworkNoiseRejection.

This repository also contains all data networks we use here, and all results of running our algorithms on those networks.

## Supporting information

**S1 Appendix. Extended results and methods.** (1) Demonstration of how the sparse WCM model better captures a data network's weight distribution. (2) Results for *k*-partite structure in real networks. (3) Discussion of using generative null models for statistical testing of data network eigenvalues. (4) Details of the unsupervised multi-way clustering algorithm. (PDF)

## Acknowledgments

We thank Adam Calhoun for permission to use his collated COSYNE submission data-set.

## Author Contributions

**Conceptualization:** Mark D. Humphries, Javier A. Caballero.

**Data curation:** Mat Evans.

**Formal analysis:** Mark D. Humphries.

**Funding acquisition:** Mark D. Humphries.

**Investigation:** Mark D. Humphries, Mat Evans, Silvia Maggi, Abhinav Singh.

**Methodology:** Mark D. Humphries, Javier A. Caballero, Silvia Maggi, Abhinav Singh.

**Software:** Mark D. Humphries, Javier A. Caballero, Silvia Maggi.

**Supervision:** Mark D. Humphries.

**Visualization:** Mark D. Humphries.

**Writing – original draft:** Mark D. Humphries.

**Writing – review & editing:** Mark D. Humphries, Javier A. Caballero, Mat Evans, Silvia Maggi, Abhinav Singh.

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
