## [Decision Letter · Decision Letter 0]

28 Apr 2021

PONE-D-20-27869

Spectral estimation for detecting low-dimensional structure in networks using arbitrary null models

PLOS ONE

Dear Dr. Humphries,

Thank you for submitting your manuscript to PLOS ONE. After careful consideration, we feel that it has merit but does not fully meet PLOS ONE’s publication criteria as it currently stands. Therefore, we invite you to submit a revised version of the manuscript that addresses the points raised during the review process.

We look forward to receiving your revised manuscript.

Kind regards,

Gabriele Oliva, Ph.D

Academic Editor

PLOS ONE

Journal Requirements:

2.Please note that PLOS ONE does not allow for the use of footnotes in its publications. As such, we ask you to remove all footnotes and move the information contained in them to the main text.

Additional Editor Comments:

Two reviews were considered, both suggesting major revision. After carefully reviewing the paper myself, I agree with the reviewers' judgement.

Reviewers' comments:

Reviewer's Responses to Questions

**Comments to the Author**

1. Is the manuscript technically sound, and do the data support the conclusions?

Reviewer #2: Yes

Reviewer #3: Yes

2. Has the statistical analysis been performed appropriately and rigorously? 

Reviewer #2: Yes

Reviewer #3: Yes

3. Have the authors made all data underlying the findings in their manuscript fully available?

Reviewer #2: Yes

Reviewer #3: Yes

4. Is the manuscript presented in an intelligible fashion and written in standard English?

Reviewer #2: Yes

Reviewer #3: Yes

5. Review Comments to the Author

Reviewer #2: Dear Authors,

I would like to thank to review your paper entitled:

Spectral estimation for detecting low-dimensional structure in networks using arbitrary null models.

The results are interesting, however before further detailed review, I would like to confirm some issues:

1. Why the authors do not put conclusion section in this manuscript? It is hard to give any comments without it. Or do the authors have any reasons for it?

2. The logical order of the manuscript is confusing. The authors have the manuscript structure as follows:

a) Introduction

b) Results

c) Discussion

d) Method

e) No conclusion

It is not easy to follow the logical order with it and to check or review it.

The paper is interesting.

Recommendation : Major Revision

Reviewer #3: The manuscript is an interesting proposal to discover low-dimensional substructures in networks using spectral graph theory. Although the key parts of the paper are of high interest both for theoretical development and applied problems, there are multiple parts of the manuscript (including its structure) that would need to be improved.

In the Introduction I miss that the authors mentioned a number of advantages (in real applications) of working with network communities rather than with the whole network. This would help to justify and motivate the proposal better.

In the Introduction, why do the authors claim that N is typically 10^2 − 10^9? While likely being true, how the authors introduce a network size is a bit awkward and they need to reword it and, perhaps, add references of real-world networks.

In the Introduction, the authors point out the disadvantages of spectral-based community detection methods. What about other community detection methods? How do they perform?

This is uncommon that a section about Results comes straight after the Introduction. However, such a Results section comprises a part of some theoretical background. I'm sure that the authors can propose a much better structure for the manuscript. Besides, a section with a discussion about the results also ahead of the section about the methodology development is also something that seems unfamiliar to the general structure.

At line 417 there is suggested an improvement about the scalability of the proposal by "a future development of more efficient memory usage". Can this point be better explained? On another matter the authors mention scalability issues for networks of size over 10^5, why this number? in which cases? how can this issue be linked to the intro ranging the network sizes in 10^2 − 10^9?

In the generation of null models, it is not clear if it is considered links weight generation in

as a process in which the links are independent to each other or an influence of the network topology may remain in the process. Another comment is about the perturbation over the link weights at line 459, since such a perturbation is not completely defined in the manuscript. Back to the question of independent variables or not, in the subsection "Node rejection" it is not clear if any hypothesis test shall be used to see if one node is part or it isn't of a low-dimensional structure. A bit more explanations would be welcome as well with respect to how the threshold proposed at line 518 works and it is justified.

The subsection of "Poisson generation links" needs a better explanation on how the weights are generated. The same comment also counts for the subsection on "practical computation".

As a general comment, the Section of Methodology shows a collection of methods, related between themselves, but not linked in a proper framework.

6. PLOS authors have the option to publish the peer review history of their article (what does this mean?). If published, this will include your full peer review and any attached files.

Reviewer #2: **Yes: **Acep Purqon

Reviewer #3: No

---

## [Author Response · Author response to Decision Letter 0]

25 May 2021

Responses to reviewers are in the uploaded "Response_to_reviews.docx" document

---

## [Decision Letter · Decision Letter 1]

21 Jun 2021

Spectral estimation for detecting low-dimensional structure in networks using arbitrary null models

PONE-D-20-27869R1

Dear Dr. Humphries,

We’re pleased to inform you that your manuscript has been judged scientifically suitable for publication and will be formally accepted for publication once it meets all outstanding technical requirements.

Kind regards,

Gabriele Oliva, Ph.D

Academic Editor

PLOS ONE

Additional Editor Comments (optional):

Both reviewers recommend acceptance. I concur with their evaluation.

Reviewers' comments:

Reviewer's Responses to Questions

**Comments to the Author**

1. If the authors have adequately addressed your comments raised in a previous round of review and you feel that this manuscript is now acceptable for publication, you may indicate that here to bypass the “Comments to the Author” section, enter your conflict of interest statement in the “Confidential to Editor” section, and submit your "Accept" recommendation.

Reviewer #2: All comments have been addressed

Reviewer #3: All comments have been addressed

2. Is the manuscript technically sound, and do the data support the conclusions?

Reviewer #2: Yes

Reviewer #3: No

3. Has the statistical analysis been performed appropriately and rigorously? 

Reviewer #2: Yes

Reviewer #3: N/A

4. Have the authors made all data underlying the findings in their manuscript fully available?

Reviewer #2: Yes

Reviewer #3: Yes

5. Is the manuscript presented in an intelligible fashion and written in standard English?

Reviewer #2: Yes

Reviewer #3: Yes

6. Review Comments to the Author

Reviewer #2: Dear Authors,

I would like to thank to review your paper entitled:

"Spectral estimation for detecting low-dimensional structure in networks using arbitrary null models"

In this paper, the authors propose a spectral approach to detect a low-dimensional structure in networks. The low-dimensional structure has useful information in community detection.

Their results show that the proposed method has good performance in both synthetic networks and real networks. Hence, it would be useful community detection method for detecting low-dimensional structure in real-world networks in many related different fields.

The overall level of the paper is interesting and well-written.

Furthermore, the paper also follows the format as required in PLOS ONE Journal.

The decision is accepted

Reviewer #3: (No Response)

7. PLOS authors have the option to publish the peer review history of their article (what does this mean?). If published, this will include your full peer review and any attached files.

Reviewer #2: **Yes: **Acep Purqon

Reviewer #3: No

---

## [Editor Report · Acceptance letter]

24 Jun 2021

PONE-D-20-27869R1 

Spectral estimation for detecting low-dimensional structure in networks using arbitrary null models 

Dear Dr. Humphries:

I'm pleased to inform you that your manuscript has been deemed suitable for publication in PLOS ONE. Congratulations! Your manuscript is now with our production department. 

Kind regards, 

on behalf of

Dr. Gabriele Oliva 

Academic Editor

PLOS ONE